# Synergistic Behavioral Response Effect of Mixtures of *Andrographis paniculata*, *Cananga odorata*, and *Vetiveria zizanioides* against *Aedes aegypti* (Diptera: Culicidae)

**DOI:** 10.3390/insects14020155

**Published:** 2023-02-03

**Authors:** Amonrat Panthawong, Jirod Nararak, Pairpailin Jhaiaun, Chutipong Sukkanon, Theeraphap Chareonviriyaphap

**Affiliations:** 1Department of Entomology, Faculty of Agriculture, Kasetsart University, Bangkok 10900, Thailand; 2Department of Medical Technology, School of Allied Health Sciences, Walailak University, Nakhon Si Thammarat 80160, Thailand; 3Excellent Center for Dengue and Community Public Health (EC for DACH), Walailak University, Nakhon Si Thammarat 80160, Thailand; 4Royal Society of Thailand, Bangkok 10300, Thailand

**Keywords:** binary mixture, repellence, *Andrographis paniculata*, *Cananga odorata*, *Vetiveria zizanioides*, excito-repellency, *Aedes aegypti*, DEET

## Abstract

**Simple Summary:**

In several Asian and Latin American nations, severe dengue is the main cause of serious disease and death. Dengue viruses are transmitted to humans via the bite of an infected *Aedes* species. Personal protection measures may be an effective method of avoiding mosquito bites. Nowadays, consumers are increasingly interested in commercial repellent products containing plant-based ingredients because they are perceived as “safe” in comparison to long-established synthetic repellents. We tested a binary plant-based mixture formulation of *Cananga odorata*, *Vetiveria zizanioides*, and the crude extract of *Andrographis paniculata* on laboratory and field strains of *Aedes aegypti* using an excito-repellency test system. The results showed that a mixture of *V. zizanioides* and *A. paniculata* at a 1:4 ratio was better than DEET. These results could lead to the further development of a combination of *V. zizanioides* and *A. paniculata* as active ingredients in a repellent that could be tested in human trials.

**Abstract:**

Each binary mixture formulation of *Vetiveria zizanioides* (L.) Nash (VZ) with *Andrographis paniculata* (Burm.f.) Wall. ex Nees (AP) or *Cananga odorata* (Lam.) Hook.f. & Thomson (CO) and AP with CO at 1:1, 1:2, 1:3, and 1:4 ratios (*v*:*v*) was investigated for behavioral responses on laboratory and field strains of *Aedes aegypti*. Irritant and repellent activities of each formulation were compared with *N*,*N*-diethyl-3-methylbenzamide (DEET) using an excito-repellency test system. The result demonstrated that the mixture of VZ:AP in all combination ratios was the most effective in inducing an irritancy response against the laboratory strain (56.57–73.33%). The highest percentage of escaped mosquitoes exposed to the mixture at a 1:4 ratio (73.33%) was significantly different from DEET (26.67%) (*p* < 0.05). Against the field strain, the strongest escape response of AP:CO at a 1:1 ratio in the contact trial (70.18%) was significantly different compared with DEET (38.33%) (*p* < 0.05). There was a weak non-contact escape pattern in all combinations of VZ:CO against the laboratory strains (6.67–31.67%). These findings could lead to the further development of VZ and AP as active ingredients in a repellent that could advance to human use trials.

## 1. Introduction

Many areas of the world are at risk from mosquito-borne diseases; in particular, the incidence of dengue has increased substantially worldwide in recent decades [1]. One model has estimated that there are 390 million dengue cases annually, with 70% of them in Asia [2]. *Aedes aegypti*, an anthropophagic daytime-biting mosquito, is the principal vector of dengue virus as well as other arboviruses [2,3]. Despite intensive efforts and various innovations to combat this species, control remains difficult, and they are highly refractory to common control measures. This has necessitated exploring methods to enhance personal protection to prevent blood-feeding and potential disease transmission.

The direct control of vectors is considered an efficient way to block the transmission cycle of pathogens, with the most common being the use of synthetic chemical insecticides [4]. Chemical compounds can protect humans from mosquito bites by either causing mortality (killing) or excito-repellency (avoidance behavior via contact irritancy and/or non-contact repellent) that interrupts or inhibits normal feeding behavior [5,6]. DEET (*N*,*N*-diethyl-3-methylbenzamide) is the standard repellent for several insects and arthropods [7]. Most commercial mosquito repellents are prepared using DEET, which may lead to health risks in the case of continuous application on exposed skin at high concentrations [8]. In addition to cost considerations, the use of insecticides carries risks associated with potential environmental contamination, reductions in non-target and beneficial organisms, and promotes the development of insecticide resistance in mosquito populations over time [9,10,11]. Therefore, it is desirable to investigate the use of natural compounds as bio-pesticides extracted from plants to protect humans from mosquito bites [4,12]. These natural product sources are generally considered safe from unwanted side effects and have become increasingly popular for protection against biting mosquitoes [12]. 

Several extracts and essential oils from botanical sources can repel *Ae. aegypti* mosquitoes, such as catnip, citronella, clove, eucalyptus, ginger, cinnamon, basil, kaffir lime, lemongrass, mountain pepper, weeping paperbark, orange, plai, vetiver, ylang-ylang, nutmeg, turmeric, and fah talai jone [13,14,15,16,17,18,19,20,21,22,23]. The repellent efficacy of essential oils can be improved by combining several essential oils from different plants, leading to a synergistic effect [24]. The synergistic use of essential oils derived from several plants has been reported to provide a higher repellent activity to *Ae. aegypti* than the individual essential oils separately [25,26]. A recent investigation studied the avoidance behavior of *Culex quinquefasciatus* Say to binary mixtures of essential oils of *Vetiveria zizanioides* or *Cananga odorata* with a crude extract of *Andrographis paniculata* using an excito-repellency (ER) test system; promising test results were obtained [27]. However, the irritancy and repellency responses of the botanical extract mixtures against *Ae. aegypti* have not been subjected to an ER assay.

Thus, the aim of this study was to use the ER test system to investigate the escape responses of medically important *Ae. aegypti* to binary mixtures of each of the essential oils of *V. zizanioides* or *C. odorata* with the crude extract of *A. paniculata* at different blending ratios. DEET was used as a positive control to compare the efficacy of each binary mixture formulation.

## 2. Materials and Methods

### 2.1. Test Mosquitoes

Laboratory and field strains of *Aedes aegypti* were used. A susceptible laboratory colony of *Ae. Aegypti* (USDA strain), originally obtained from the United States Department of Agriculture (USDA), has been maintained for at least 20 years at the Department of Entomology, Faculty of Agriculture, Kasetsart University, Bangkok, Thailand and is completely susceptible to all commonly used insecticides [17]. For the field strain, larvae and pupae were collected from Pu Teuy village, Sai Yok district, Kanchanaburi province, Thailand (14.3376268 N, 98.9817521 E). Both adult mosquito strains were provided with cotton pads soaked with 10% sucrose solution from the day of emergence. They were maintained in screened cages (30 cm × 30 cm × 30 cm) under uniformed climatic conditions of 25 ± 2 °C, 80 ± 10% relative humidity, and a 12:12 h (light:dark) cycle. Females were free-mated and permitted to feed on blood in an artificial membrane blood-feeding system [28] on the fourth day after emergence. Gravid females then laid their eggs within two days following blood feeding. A small plastic cup containing moist filter paper (10 cm in diameter) was provided in the cages as an oviposition container. The eggs in the oviposition dishes were initially dried (‘conditioned’) at room temperature for approximately 24–48 h. The eggs that had been affixed to the filter papers were immersed in fresh water in hatching trays. One to two days after hatching, approximately 250 larvae were transferred to individual plastic trays (20 × 30 × 5 cm) containing 1500 mL of tap water and provided with 5–6 fish granules. The pupae were transferred to emergence cups and placed in screened cages. Adults were provided with 10% sugar meal ad libitum until 12 h prior to the assay. All procedures were approved under the Animal Use Protocol entitled “Synergistic repellent and irritant effect of the binary mixture of ylang-ylang, fah talai jone, and vetiver oil against *Aedes aegypti* (Diptera: Culicidae) (U1-09518-2564)” to Miss Amonrat Panthawong by the Kasetsart University Institutional Animal Care and Use Committee, Bangkok, Thailand.

### 2.2. Test Compounds 

The essential oils of ylang-ylang, *Cananga odorata* (CO) (Batch No. 20010111-1), and vetiver, *Vetiveria zizanioides* (VZ) (Batch No. 200402491), and the crude extract (100% purity) of fah talai jone, *Andrographis paniculata* (AP) (Batch No. 19121204-1), used in this study were purchased from the Thai-China Flavours and Fragrances (TCFF) Industry Co., Ltd., Ayutthaya, Thailand. The essential oils were obtained from selected plant parts using a steam distillation method, and their chemical composition was analyzed based on gas chromatography and mass spectrometry. The CO essential oil (CO EO) was extracted from flowers, while the VZ essential oil (VZ EO) was from the roots. The main components of CO EO were β-caryophyllene and linalool [22], while in VZ EO, they were β-vetivone and khusimol [29]. The AP crude extract (AP CE) was obtained by maceration of *A. paniculata* leaves containing andrographolide and 14-deoxyandrographolide in ethanal [21]. Each test compound was prepared with absolute ethanol (Merck, Darmstadt, Germany) to obtain 2.5% (*v*/*v*). This concentration was selected based on other publications [17,21] reporting the highest mosquito-repellent potential. Binary mixture formulations were prepared of VZ EO with AP CE or CO EO and of AP CE with CO EO at ratios of 1:1, 1:2, 1:3, and 1:4 (*v*/*v*). The VZ EO was purchased from TCFF along with the main chemical profile document. We used the same batch of VZ EO for our experiments.

### 2.3. Paper Impregnation 

Whatman No. 1 filter papers were cut into rectangles of size 14.7 cm × 17.5 cm. Separate papers were treated with each binary mixture formulation using a calibrated micropipette delivering 2.8 mL of solution. Untreated control papers were treated with absolute ethanol only. DEET (2.5% *v*/*v*) was used as a gold-standard repellent. The treated papers were placed on a plastic sheet and air-dried at room temperature for at least 1 h before use. Multiple papers in each formulation were discarded after being used once [17,21]. 

### 2.4. Behavioral Assay

The excito-repellency assay system has been described by Chareonviriyaphap et al. [30] and Roberts et al. [31]. Briefly, a set of 4 test chambers was used simultaneously to evaluate the different types of contact irritant and non-contact spatial repellents, consisting of 2 treatment chambers (either contact or non-contact) and 2 paired control chambers, respectively. The testing chamber is connected to the external receiving box for escape response observation. In the contact chamber, mosquitoes were allowed to make direct physical contact with the treated paper, whereas treated papers were placed behind a mesh screen preventing direct tarsal contact with test mosquitoes. To begin testing, 15 active female mosquitoes, non-blood fed and 3–5 days old, were introduced into each testing chamber using an entomological mouth aspirator. Mosquitoes were allowed to briefly (3 min) adjust to the environmental conditions inside the test chamber before the exit portal was opened, leading to the receiving box. The number of escaping mosquitoes was recorded every minute for 30 min. At the end of the exposure period, escaped and non-escaped (those remaining in the chamber) mosquitoes were transferred to individual containers and provided 10% sugar solution during the holding period. Knockdown response (moribund mosquitoes, unable to fly) was recorded immediately after 30 min, and final mortality was recorded after 24 h in both treatments and controls for escaped and non-escaped mosquitoes. Each binary mixture formulation was tested in 4 replicates (n = 60) during daylight hours between 08:00 and 16:30 h. The test chambers were carefully cleaned with 95% ethyl alcohol and allowed to dry for at least 24 h before beginning the next experiment.

### 2.5. Data Analysis

Abbott’s formula [32] was used to adjust the escape responses if the paired control escape number was between 5–20%. The number of knockdown and dead mosquitoes was used to calculate initial knockdown and final mortality responses. Using Kaplan–Meier survival analysis [31], the mean percentages of escape response and mortality for each test chamber and mixture formation were analyzed. As previously described, the design of the contact chamber offered the determination of a combined contact excitation (‘irritancy’) and non-contact repellency. To estimate the true irritancy, the escape percentage was therefore adjusted again by comparing the numbers of mosquitoes that successfully escaped in each paired contact and non-contact trial [19] using the following equation: (1 − [number of contacts in test × number of non-contact escapes/number of non-contacts in test × number of contact escapes]) × 100. This measurement, a reciprocal of the Henderson–Tilton (H–T) formula designed to measure the effects of toxic chemicals on populations, enables the adjustment of the ‘crude’ contact escape percentage to provide an estimate of the escape percentage due to contact irritancy to exclude those effects due to repellency [33]. Then, the adjusted contact escape divided by the pre-adjusted (combined effects) escape percentage was calculated, resulting in an estimation of the ‘percentage effect’ due to contact excitation alone. For computational purposes, escaped mosquitoes were designated as “dead”, while non-escaped mosquitoes were defined as “survivors”. Survival analysis was then used to compare the escape responses between contact and non-contact trials (using initial escape data). A log-rank method [34] was used to compare the patterns of escape behavior. The escape time (ET) in minutes for 25% (ET_25_), 50% (ET_50_), and 75% (ET_75_) of mosquitoes to escape was calculated for each formulation. The escape response data were analyzed using the SAS software version 9 (SAS Institute, Cary, NC, USA). Statistical significance for all tests was tested at 5% (*p* < 0.05).

## 3. Results

The contact irritancy and non-contact repellency effects of binary mixtures at different blending ratios of VZ EO, AP CE, and CO EO compared with DEET on laboratory and field strains of *Ae. aegypti* are shown in Table 1 and Figure 1, Figure 2 and Figure 3. The escape rates represent the probabilities of mosquitoes escaping from a chamber with a particular chemical or mixture of plant-based combinations. A strong escape pattern was observed with VZ EO:AP CE in the contact trial for all combinations (56–73%) in the laboratory strain as well as all combinations of AP CE:CO EO for all combinations (42–70%) of the field strain. The combination ratio of 1:4 produced the highest percentage of escaped mosquitoes of 73.33% for VZ EO:AP CE from the contact trial. Overall, AP CE:CO EO produced more robust escape responses than VZ EO:AP CE and VZ EO:CO EO in the contact and non-contact trials. With VZ EO:CO EO, there was a weak non-contact escape pattern in all combinations of the laboratory strains (6–31%).

After a ‘true contact irritancy’ adjustment, the percentage of contact escape response decreased, except the laboratory strain exhibited 0% escape in the ‘true’ contact response for AP CE:CO EO (1:4).

Multiple log-rank comparisons between any two combinations of exposure to repellent compounds in either contact or non-contact trials are shown in Table 2. For AP CE:CO EO, there were significant differences in escape responses when 1:1 was compared to the other combinations in the contact trials but not at 1:1 compared with the ratios of 1:2 and 1:3 in the non-contact trials. For VZ EO:AP CE, there were significant differences in the escape responses between combinations only for 1:1 compared to 1:3 in the non-contact trial of the laboratory strain. Against the laboratory strain, the irritant response due to all combination ratios of VZ EO:AP CE was not significantly different from DEET in non-contact trials (*p* > 0.05) but was significantly different in contact trials. For the field strain, all mixtures of VZ EO:AP CE were not significantly different from that elicited by DEET alone (*p* > 0.05), except for VZ EO:AP CE in the contact trial. Multiple log-rank comparisons between the contact and non-contact trials for each mixture were conducted (Table 3). Notably, the VZ EO:AP CE responses were all significantly different in the escape patterns for the *Ae. aegypti* laboratory strain in all paired comparisons between the contact and non-contact trials. For AP CE:CO EO, there were significant differences in the escape patterns in the paired comparisons between the contact and non-contact trials, except for the 1:2 combination. 

The escape patterns from chambers treated with test compounds were defined based on the escape time for 25% (ET_25_), 50% (ET_50_), and 75% (ET_75_) of the test population to leave the treated chamber in 30 min. The ET_25_ laboratory and field strains in the contact trial had the most rapid response to VZ EO:AP CE for all combinations within 1–2 min. VZ EO:CO EO (1:2) produced delayed escape responses with E_25_ values in both laboratory (28 min) and field (22 min) strains. No response was observed for the ET_75_ value in both the contact and non-contact trials for the laboratory and field strains with all tested compounds. There was a faster escape response in the contact trial to all combinations compared to the similar non-contact trial. There were only ET_50_ values in the contact trial of VZ EO:AP CE for the laboratory strain and of VZ EO:CO EO and AP CE:CO EO in the field strain (Table 4).

Based on the Kaplan–Meier survival analysis, compared with other test condition, the strongest escape patterns were observed for laboratory and field strains exposed to AP CE:CO EO (1:1) in the contact trial. VZ EO:AP CE (1:4) had the highest escape pattern response (>70% in 10 min) for the laboratory strain in the contact trial, whereas the field strain’s escape response was less than 50% for all combinations. VZ EO:CO EO (1:4) produced an escape response of almost 70% for the field strain compared with less than 50% for the laboratory strain during the contact test. 

Knockdown (KD) of escaped and non-escaped mosquitoes was observed during the exposure period (30 min). The KD percentage for the non-escaped field strain in the contact trials for AP CE:CO EO (1:1) was 27.50%. Likewise, there was a marked mortality percentage (27.50%) for the escaped field strain in the contact trials for AP CE:CO EO (1:1).

## 4. Discussion

The current study measured the irritant contact and non-contact repellent characteristics of binary mixtures of VZ EO with AP CE or CO EO, and of AP CE with CO EO for ratios of 1:1, 1:2, 1:3, and 1:4 (*v*:*v*) against *Ae. aegypti* using the excito-repellency test system. The results showed that the binary mixture of VZ EO:AP CE in all proportions was the most effective in inducing an irritancy response against the laboratory strain (68–73% escape) and was significantly higher than DEET (26% escape). Similar to the previous study, Boonyuan et al. [27] reported that the greatest contact irritant of VZ EO:AP CE at a ratio of 1:1 (*v*:*v*) against both laboratory and field strains of *Cx. Quinquefasciatus* (96.67% escape) was significantly greater than DEET (88.67% escape). This difference percentage of escape may depend on the mosquitoes tested. *Culex* mosquitoes are more sensitive to repellents than *Aedes* and *Anopheles* mosquitoes, which are repellent-tolerant species [7]. DEET showed >90% repellent against *Ae. aegypti* at 19 and 25% concentrations [7], while 2.5% DEET was used in this study. For the field strain, a strong escape response was observed for AP CE:CO EO for all proportions in the contact trial (42–70% escape). However, the highest percentage of escaped mosquitoes exposed to VZ EO:AP CE was at a ratio of 1:4 (73.33%), while for AP CE:CO EO, it was at a 1:1 ratio (70.18%). This difference could have originated from the different types of the synergistic chemical composition of each essential oil combination.

VZ EO consists of a complex mixture of more than 200 compounds [35]. The major components of VZ EO are β-vetivone, khusimol, α-vetivone, vitiveryl acetat, vetiverol, vitivone, and terpenes [27,36]. VZ EO has been reported to repel several insects, such as termites, mosquitoes, weevils, and beetles [12,37,38]. Recently, Nararak et al. [35] demonstrated that VZ EO and two of their constituents (valencene and vetiverol) had potential as active ingredients for repelling *Ae. Aegypti*. Furthermore, another study showed that the combination of vetiver (10%), citronella (20%), and hairy basil (10%) oils in nanoemulsion promoted a longer protection time (>3 h) against *Ae. aegypti* [39]. For AP CE, Sukkanon et al. [21] showed that andrographolide and 14-deoxyandrographolide were the major diterpenoids found in *A. paniculata* plants based on high-performance liquid chromatography-mass spectrometry. They also found that AP CE exhibited strong non-contact repellency against *Ae. aegypti* (72% escape). Edwin et al. [40] reported that andrographolide at a concentration of 12 ppm exhibited an effective repellency percentage of >90% with a protection time of 15–210 min compared to the *Ae. aegypti* control group. The results of the current study also indicated that the laboratory strain of *Ae. aegypti* produced the lowest response to all combinations of VZ EO:CO EO in the non-contact trail. Consistent with the previous research, 10% CO EO provides the strongest value against *Ae. aegypti*, with only 66% repellency [41]. However, Sukkanon et al. [22] reported that CO EO at 2.5% concentration caused *Ae. aegypti* to escape at a rate of 39% in the non-contact repellency trial and 59% in the contact irritancy trial. The use of CO EO alone may be more effective in repelling *Ae. aegypti* than the combination with VZ EO. The repellent action of the plant extracts tested varied depending on the plant part, the solvent used in extraction, and the dose [42]. 

The estimation of escape time indicated that VZ EO:AP CE had the strongest excitation and repellency actions with both the laboratory and field strains of *Ae. aegypti* in the contact trial. Most plant essential oils are volatile and act on mosquitoes in the volatile phase. Therefore, their effectiveness and protection time are time-limited [38,43]. Improved repellency of plant-derived topical repellents has been shown after formulation with some bases or fixative materials, such as liquid paraffin, vanillin, and salicyluric acid [44,45,46,47]. According to Chareonviriyaphap et al. [48], three laboratory strains utilized in this study had been continuously colonized for 15–20 years and appeared to have lost some of their natural behavioral avoidance response to deltamethrin. Two colonized populations of *Anopheles albimanus* Wiedemann from Panama and El Salvador also exhibited suppressed avoidance responses. Both populations had been maintained in laboratories for more than two decades and exhibited significantly less avoidance of insecticide exposure than populations captured in the wild. A long-term (15 years) colonized strain of *Anopheles dirus* Peyton and Harrison exhibited a lower escape response to deltamethrin than other strains from Thailand [49]. Before extrapolating results to “normal” mosquito behavior in the wild, it is important to think carefully about how to use mosquito populations that have been there for a long time.

The prevention and control of most vector-borne diseases (those lacking effective vaccines, for example) remain dependent on various vector control strategies to decrease the transmission risk. In many instances, this requires the use of chemical insecticides as outdoor space spray and indoor residual spray applications and insecticide-treated bed nets to prevent adult mosquito blood feeding [5,50,51]. After decades of insecticide exposure, the development of resistance by mosquitoes to synthetic compounds has reduced the effectiveness of traditional chemical-based prevention and control methods [10,52,53,54]. For this reason, the potential use of plant-derived products as a source of various extracts and essential oils as natural repellents is an attractive option to replace or supplement synthetic compounds for the control of vectors and prevention of disease transmission. 

Several plant-based essential oils have been evaluated for their insect-repellent activity for protection against mosquitoes and other arthropod pests in Thailand. These have included *Ocimum* spp. [46,55], *Nepeta cataria* [14], *Citrus hystrix* [56], *Melaleuca leucadendron*, *Litsea cubeba*, and *Litsea salicifolia* [13], *Citrus aurantium*, *Cinnamomum verum*, *Cymbopogon winterianus*, *Syzygium aromaticum* [15], *V. zizanioides* [12,17], *C. odorata* [57,58], and *A. paniculata* [21]. Notably, several research studies have demonstrated that mixtures of essential oils provide a higher repellency against *Ae. aegypti* than a single essential oil [59,60,61], probably by increasing the effectiveness of the number of chemicals in the mixture [25,26,61] and synergism between the constituent compounds [27]. From these data, essential oil mixtures could be a cheaper alternative for human populations and promising tools such as mosquito repellents [61].

## 5. Conclusions

In summary, VZ EO:AP CE at a 1:4 ratio in the contact trial displayed the significantly strongest escape patterns in the laboratory strain compared to the other test conditions, while AP CE:CO EO at a 1:1 ratio demonstrated strong irritant activity against the field strain. The laboratory strain produced the lowest response to the mixture of VZ EO:CO EO, especially in the non-contact trial. However, the promising attributes of essential oils of *V. zizanioides* and *C. odorata* and of the crude extract of *A. paniculata* are that they are derived from natural sources and are relatively less toxic compared to synthetic repellents. Therefore, more work is needed to optimize these botanicals as potential commercial mosquito repellents, not only for topical use but for possible inclusion in other control tools.

## Figures and Tables

**Figure 1 insects-14-00155-f001:**
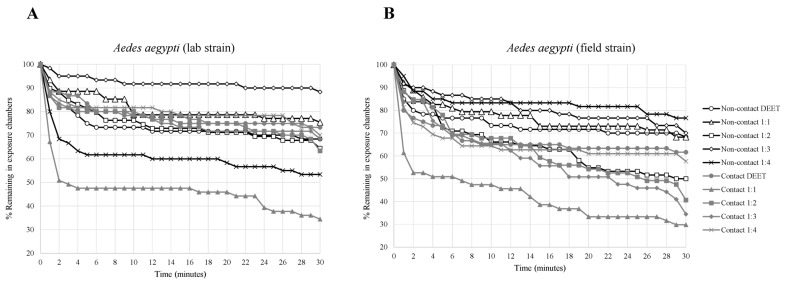
Analysis of survival for *Aedes aegypti* strains in treated non-contact and contact ER tests ((**A**) laboratory strain, (**B**) field strain). Escape responses were noted each minute for 30 min exposure to different binary mixture formulations of 2.5% AP CE:2.5% CO EO. Paired control escape responses not shown.

**Figure 2 insects-14-00155-f002:**
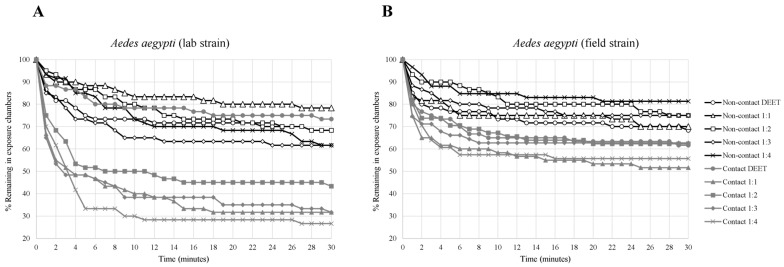
Analysis of survival for *Aedes aegypti* strains in treated non-contact and contact ER tests ((**A**) laboratory strain, (**B**) field strain). Escape responses were noted each minute for 30 min exposure to different binary mixture formulations of 2.5% VZ EO:2.5% AP CE. Paired control escape responses not shown.

**Figure 3 insects-14-00155-f003:**
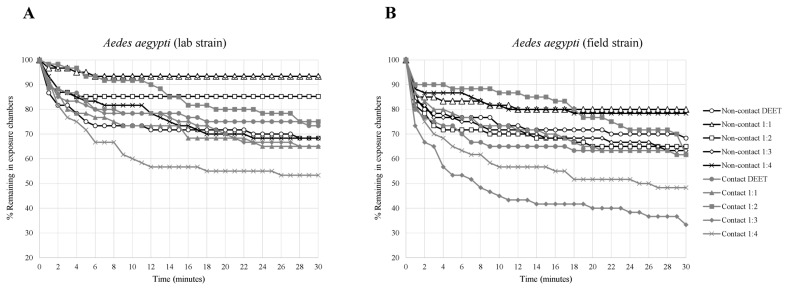
Analysis of survival for *Aedes aegypti* strains in treated non-contact and contact ER tests ((**A**) laboratory strain, (**B**) field strain). Escape responses were noted each minute for 30 min exposure to different binary mixture formulations of 2.5% VZ EO:2.5% CO EO. Paired control escape responses not shown.

**Table 1 insects-14-00155-t001:** Avoidance response of *Aedes aegypti* exposed to each binary mixture of AP CE:CO EO, VZ EO:AP CE, and VZ EO:CO EO at different blending ratios and DEET in contact and non-contact trials.

Repellents	ER Assay	*Aedes aegypti* (Lab Strain)					*Aedes aegypti* (Field Strain)				
		Treatment			Control		Treatment			Control	
		N	%ESC ^a^	%ESC ^b,c^	N	%ESC	N	%ESC ^a^	%ESC ^b,c^	N	%ESC
DEET	C	60	26.67	24.14 (0)	60	3.33	60	38.33	37.28 (6.67)	60	1.67
	NC	60	31.67	30.51	60	1.67	60	31.67	29.32	60	3.33
AP CE:CO EO (1:1)	C	61	65.57	65.00 (40.98)	61	1.64	57	70.18	69.17 (38.43)	61	3.28
	NC	61	24.59	24.59	60	0	63	31.75	29.47	62	3.23
AP CE:CO EO (1:2)	C	60	36.67	35.61 (1.07)	61	1.64	59	59.32	58.63 (9.32)	60	1.67
	NC	59	35.59	35.59	60	0	62	50.00	49.15	60	1.67
AP CE:CO EO (1:3)	C	60	31.67	29.32 (20.00)	60	3.33	61	65.57	64.38 (35.57)	60	3.33
	NC	60	11.67	8.63	60	3.33	60	30.00	27.59	60	3.33
AP CE:CO EO (1:4)	C	60	30.00	27.59 (0)	60	3.33	60	42.37	40.42 (19.04)	60	3.28
	NC	60	46.67	43.86	60	5.00	60	23.33	20.69	60	3.33
VZ EO:AP CE (1:1)	C	60	68.33	67.79 (46.67)	60	1.67	60	48.33	46.55 (18.33)	60	3.33
	NC	60	21.67	18.97	60	3.33	60	30.00	27.59	60	3.33
VZ EO:AP CE (1:2)	C	60	56.57	55.83 (25.00)	60	1.67	61	37.70	35.55 (12.70)	60	3.33
	NC	60	31.67	30.51	60	1.67	60	25.00	22.42	60	3.33
VZ EO:AP CE (1:3)	C	60	68.33	68.33 (30.00)	60	0	59	37.29	35.13 (12.29)	60	3.33
	NC	60	38.33	38.33	60	0	60	25.00	23.73	60	1.67
VZ EO:AP CE (1:4)	C	60	73.33	72.41 (35.00)	60	3.33	61	44.26	42.34 (25.62)	60	3.33
	NC	60	38.33	37.28	60	1.67	59	18.64	15.84	60	3.33
VZ EO:CO EO (1:1)	C	60	35.00	35.00 (28.33)	60	0	60	38.33	38.33 (18.33)	60	0
	NC	60	6.67	6.67	60	0	60	20.00	17.24	60	3.33
VZ EO:CO EO (1:2)	C	60	25.00	25.00 (10.25)	60	0	60	38.33	37.28 (3.33)	60	1.67
	NC	61	14.75	14.75	60	0	60	35.00	32.76	60	3.33
VZ EO:CO EO (1:3)	C	60	35.00	32.76 (28.33)	60	3.33	60	66.67	65.52 (30.00)	60	3.33
	NC	60	6.67	6.67	60	0	60	36.67	34.49	60	3.33
VZ EO:CO EO (1:4)	C	60	46.67	46.67 (15.00)	60	0	60	51.67	50.85 (30.00)	60	1.67
	NC	60	31.67	31.67	60	0	60	21.67	20.34	60	1.67

AP CE, *Andrographis paniculata* crude extract; CO EO, *Cananga odorata* essential oil; VZ EO, *Vetiveria zizanioides* essential oil; Esc, escaped mosquitoes; N, number of tested mosquitoes; C, contact; NC, non-contact. ^a^ Percent escape response before adjustment with paired control response. ^b^ Adjusted escape rate based on paired control response using Abbott’s formula. ^c^ Adjusted contact escape percent based on paired non-contact escape using Henderson–Tilton’s formula.

**Table 2 insects-14-00155-t002:** Log-rank comparisons of escape responses between each binary mixture formulation at different blending ratios and DEET against each strain of *Aedes aegypti* in contact and non-contact trials.

Plant Extract	Blending Ratio	*p*-Value			
		*Aedes aegypti* (Lab Strain)		*Aedes aegypti* (Field Strain)	
		Contact	Non-Contact	Contact	Non-Contact
AP CE:CO EO	1:1 vs. 1:2	0.0006 *	0.1943	0.0508	0.0468 *
	1:1 vs. 1:3	0.0001 *	0.0659	0.1784	0.7732
	1:1 vs. 1:4	<0.0001 *	0.0076 *	0.0024 *	0.3069
	1:2 vs. 1:3	0.6204	0.0021 *	0.4854	0.0209 *
	1:2 vs. 1:4	0.4501	0.1500	0.1858	0.0029 *
	1:3 vs. 1:4	0.8103	<0.0001 *	0.0474 *	0.4469
	1:1 vs. DEET	<0.0001 *	0.3310	<0.0001 *	0.9017
	1:2 vs. DEET	0.2818	0.7618	0.0040 *	0.0707
	1:3 vs. DEET	0.5646	0.0064 *	0.0004 *	0.7202
	1:4 vs. DEET	0.7181	0.0874	0.1167	0.2688
VZ EO:AP CE	1:1 vs. 1:2	0.1817	0.2414	0.2350	0.4681
	1:1 vs. 1:3	0.9985	0.0386 *	0.2744	0.5532
	1:1 vs. 1:4	0.5837	0.0581	0.6656	0.1402
	1:2 vs. 1:3	0.1724	0.3195	0.9421	0.9118
	1:2 vs. 1:4	0.0635	0.4635	0.4395	0.4340
	1:3 vs. 1:4	0.6305	0.7581	0.4944	0.3800
	1:1 vs. DEET	<0.0001 *	0.1924	0.0242 *	0.8452
	1:2 vs. DEET	0.0005 *	0.8442	0.2622	0.3634
	1:3 vs. DEET	<0.0001 *	0.4646	0.2378	0.4246
	1:4 vs. DEET	<0.0001 *	0.6120	0.0641	0.0916
VZ EO:CO EO	1:1 vs. 1:2	0.1561	0.1472	0.7262	0.0726
	1:1 vs. 1:3	0.9549	0.9952	0.0014 *	0.0523
	1:1 vs. 1:4	0.1989	0.0006 *	0.1230	0.8425
	1:2 vs. 1:3	0.1824	0.1423	0.0002 *	0.9147
	1:2 vs. 1:4	0.0044 *	0.0410 *	0.0484 *	0.1046
	1:3 vs. 1:4	0.1622	0.0006 *	0.1042	0.0783
	1:1 vs. DEET	0.3355	0.0005 *	0.2941	0.1630
	1:2 vs. DEET	0.6993	0.0342 *	0.3975	0.6757
	1:3 vs. DEET	0.3678	0.0005 *	<0.0001 *	0.5911
	1:4 vs. DEET	0.0267 *	0.8908	0.0148 *	0.2241

* Indicates significantly different at *p* < 0.05; AP CE, *Andrographis paniculata* crude extract; CO EO, *Cananga odorata* essential oil; VZ EO, *Vetiveria zizanioides* essential oil.

**Table 3 insects-14-00155-t003:** Log-rank comparisons of escape responses between contact and non-contact trials.

Plant Extract	Blending Ratio	*Aedes aegypti*	
		Lab Strain	Field Strain
AP CE:CO EO	1:1	<0.0001 *	<0.0001 *
	1:2	0.9268	0.4452
	1:3	0.0072 *	0.0001 *
	1:4	0.0450 *	0.0215 *
VZ EO:AP CE	1:1	<0.0001 *	0.0364 *
	1:2	0.0017 *	0.0961
	1:3	0.0006 *	0.1262
	1:4	<0.0001 *	0.0016 *
VZ EO:CO EO	1:1	0.0002 *	0.0379 *
	1:2	0.2265	0.9940
	1:3	0.0002 *	0.0011 *
	1:4	0.0730	0.0008 *

* Indicates significantly different at *p* < 0.05; AP CE, *Andrographis paniculata* crude extract; CO EO, *Cananga odorata* essential oil; VZ EO, *Vetiveria zizanioides* essential oil.

**Table 4 insects-14-00155-t004:** Estimated escape time in minutes for 25% (ET_25_) and 50% (ET_50_) of mosquitoes to exit contact and non-contact test chambers exposed to test compounds.

Compound	Test	Ratio	*Ae. aegypti* (Lab Strain)		*Ae. aegypti* (Field Strain)	
			ET_25_	ET_50_	ET_25_	ET_50_
AP CE:CO EO	Contact	1:1	1	3	1	7
		1:2	15	-	6	26
		1:3	13	-	5	23
		1:4	27	-	2	-
	Non-contact	1:1	-	-	15	-
		1:2	11	-	5	29
		1:3	-	-	27	-
		1:4	2	-	-	-
VZ EO:AP CE	Contact	1:1	1	4	1	-
		1:2	1	7	2	-
		1:3	1	3	1	-
		1:4	1	4	2	-
	Non-contact	1:1	-	-	6	-
		1:2	13	-	28	-
		1:3	4	-	17	-
		1:4	10	-	-	-
VZ EO:CO EO	Contact	1:1	8	-	8	-
		1:2	28	-	22	-
		1:3	15	-	1	8
		1:4	4	-	2	25
	Non-contact	1:1	-	-	-	-
		1:2	-	-	3	-
		1:3	-	-	6	-
		1:4	14	-	-	-
DEET	Contact		18	-	3	-
	Non-contact		5	-	10	-

AP CE, *Andrographis paniculata* crude extract; CO EO, *Cananga odorata* essential oil; VZ EO, *Vetiveria zizanioides* essential oil.

## Data Availability

All relevant data are included in the article.

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
