# Peer review of "Synergistic Behavioral Response Effect of Mixtures of Andrographis paniculata, Cananga odorata, and Vetiveria zizanioides against Aedes aegypti (Diptera: Culicidae)"

_insects, 2023, doi:10.3390/insects14020155_

Round 1
Reviewer 1 Report
This study investigates the repellent effects of various combinations of plant-derived essential oils/extracts on lab and field strains of Aedes aegypti. At the concentrations used, the authors find that some of these show a significantly greater repellent effect than the gold standard repellent DEET, suggesting these plant-derived chemicals could be further developed as cheaper and less toxic repellents against medically important mosquitoes.
The paper overall is nicely written with clear Introduction and Methods. Results could benefit from including more data from DEET experiments for comparison.
Abstract
I suggest that EO and CE be removed from the abstract to improve the readability. As these are not defined in the abstract it is confusing to have these extra letters. For the purposes of the abstract it is sufficient to just write VZ:AP or AP:CO etc. Alternatively EO and CE should be defined in abstract.
Results
If the experiments were done, the Kaplan-Meier curve (Figure 1-3) and estimated escape time (Table 4) should also be presented for the DEET experiment for comparison.
Discussion
It should be noted that the DEET concentrations of 2.5% used in this study are about 10x less than recommended in repellents e.g. the US CDC recommends >20% DEET in insect repellents.
Reviewer 2 Report
Comments:
The authors investigated the effect of mixtures of Andrographis paniculata, Cananga odorata, and Vetieria zizanioides against Aedes aegypti. In this research, the authors compared the laboratory and field strans with dfferent blending ratio of binary mixture formulation. They found VZ EO: AP CE at a 1:4 ratio in the contact trial displayed the significantly strongest excape patterns of laboratory strain compared to the other conditions, while AP CE: CO EO at a 1:1 ratio demonstrated strong irritant activity against the field stain. The analysis results will facilitate in better understanding of the mixture of botanical repellent. However, I have some comments about the design.
Major comments
1. In the manuscipt, the authors mentioned the laboratory strain and the field strain. The define difference between the two strain does not described in the design. Does the field strain have strong resistence?
2. As authors mentioned, VZ EO constists of a complex mixture of more than 200 compounds. In the methods, the extraction process does not presented for these mixture. How can the mixture be kept the stable component in the repeat experiments?
Minor comments
1. Line104, please explain or change “A. Panthawong”.
2. Line 181, “ET” is the first appearance. Please give the total spelling.
3. Line 494, reference “Suwansirisilp, K., S.....”should return.
4. Line 385-514, the references format should be changed according to the request of Insects.
Round 2
Reviewer 2 Report
Accept.